# Optimization of Canolol Production from Canola Meal Using Microwave Digestion as a Pre-Treatment Method

**DOI:** 10.3390/foods12020318

**Published:** 2023-01-09

**Authors:** Ruchira Nandasiri, Olamide Fadairo, Thu Nguyen, Erika Zago, M. U. Mohamed Anas, N. A. Michael Eskin

**Affiliations:** 1Department of Food & Human Nutritional Sciences, University of Manitoba, Winnipeg, MB R3T 2N2, Canada; 2Richardson Centre for Functional Foods & Nutraceuticals, 196 Innovation Drive, Winnipeg, MB R3T 6C5, Canada; 3St. Boniface Hospital Albrechtsen Research Centre, 351 Tache Avenue, Winnipeg, MB R2H 2A6, Canada; 4Deptartment of Biological Sceinces, University of Alberta, Edmonton, AB T6G 2E9, Canada

**Keywords:** microwave-assisted extraction (MAE), canolol, sinapine, high temperature, de-oiled canola, processing

## Abstract

Canola meal, the by-product of canola oil refining, is a rich source of phenolic compounds and protein. The meal, however, is primarily utilized as animal feed but represents an invaluable source of nutraceuticals. Of particular interest are the sinapates, sinapine and sinapic acid, with the decarboxylation of the latter to form canolol. Extracting these phenolics has been carried out using a variety of different methods, although there is an urgent need for environmentally safe and sustainable methods. Microwave-assisted solvent extraction (MAE), as a green extraction method, is receiving considerable interest. Its ease of use makes MAE one of the best methods for studying multiple solvents. The formation of canolol, from sinapine and sinapic acid, is primarily dependent on temperature, which favors the decarboxylation reaction. The application of MAE, using the Multiwave^TM^ 500 microwave system with green extractants, was undertaken to assess its ability to enhance the yield of sinapates and canolol. This study examined the effects of different pre-treatment temperature-time combinations of 140, 150, 160, and 170 °C for 5, 10, 15, 20, and 30 min on the extraction of canolol and other canola endogenous phenolic compounds. Total phenolic content (TPC), total flavonoid content (TFC), as well as metal ion chelation (MIC) and DPPH radical activity of the different extracts were assessed. The results confirmed that extractability of canolol was optimized with methanol at 151 °C and with ethanol at 170 °C with pre-treatment times of 15.43 min and 19.31 min, respectively. Furthermore, there was a strong positive correlation between TPC and TFC (*p* < 0.05) and a negative correlation between TFC and DPPH radical activity. Interestingly, no significant correlation was observed between MIC and DPPH. These results confirmed the effectiveness of MAE, using the novel Multiwave^TM^ 500 microwave instrument, to enhance the yield of canolol. This was accompanied by substantial improvements in the antioxidant activity of the different extracts and further established the efficacy of the current MAE method for isolating important natural phenolic derivatives for utilization by the nutraceutical industry.

## 1. Introduction

Green chemistry and its associated technologies have gained considerable attention in recent years by both the Federal and Provincial governments of North America. The Canadian government favors green technology and its associated applications for industrial use [1]. In the 2022 section of Chapter 3 report dealing with clean air and a strong economy, the Canadian government has prioritized green technology as the future direction for industries and associated organizations [1]. This technology can be applied to the oilseed industry to reduce harmful the use of harmful chemicals and extraction solvents, and to minimize the associated detrimental environmental effects. Conventional processing techniques utilize large amounts of solvents; for instance, 1 g of substrate (meal) requires 70 mL ethanol, which is considered uneconomical and environmentally undesirable [2,3].

During oil refining, the majority of the phenolic compounds are retained in the meal fraction [4]. Hence, the meal is a rich source of phenolic compounds, particularly sinapate derivatives [5]. Sinapine is the choline ester of sinapic acid, is the most abundant phenolic compound present in the meal, and accounts for 80% of the total phenols present [6]. It has been reported that sinapine undergoes structural changes in which the choline ester is first removed by hydrolysis to form sinapic acid. This is followed by decarboxylation of sinapic acid with the formation of canolol (2,6-dimethoxy-4-vinylphenol), a potent antioxidant compound [5,7]. Khattab et al. [7] and Terpinc et al. [8] both highlighted canolol’s powerful radical scavenging properties, which were shown to protect lipids and proteins from oxidation [9,10]. In addition to the major sinapates, other thermo-generative phenolic compounds including canolol [5] and its derivatives have been shown to exhibit greater antioxidative properties during thermal processing [11,12]. The powerful antioxidant properties of canolol have also been shown to be responsible for its strong anticancer properties [13].

The application of more energy efficient processing combined with a reduction in harmful solvents/ingredients has become an important concern by industry in recent years. Novel energy efficient methods have been introduced to reduce the environmental impact by creating sustainable processing techniques. In addition, hexane, the primary solvent used by the refining industry, is being eliminated to minimize its presence in the residual oils and pressed cake [14]. At present, de-solventization of hexane from the meal is critical for the production of safe meal for animal feed and the nutraceutical industries [15]. Microwave-assisted solvent extraction (MAE) is a novel green technique that has gained considerable attention by the food and pharmaceutical industries due to its ease of use, high efficiency, and higher yields [16,17]. However, its application in the oilseed industry has been limited due to its high costs. Nevertheless, the new green economy initiative of the Canadian government is encouraging the use of such techniques as they are more energy efficient than the current commercial solvent extraction methods [1].

The targeted co-extraction of canola phenolic compounds, particularly canolol, using the MAE is an attractive alternative for producing value-added products as sources of nutraceuticals [18]. Khattab et al. [7] successfully demonstrated the formation of canolol using a microwave oven.

As a solvent extraction technique, however, microwave have not been fully evaluated for extracting canolol and other thermo-generative compounds. MAE has many advantages, including reducing the surface tension and viscosity of the extracting solvents at higher extraction temperatures. Consequently, this enhances the solubility and mass transfer of targeted phenolics, including canolol and other thermo-generative compounds [18]. This study targets the extraction of canolol and sinapate derivatives from canola meal using a relatively new microwave system as a way to substantially increase their yield. The effectiveness of MAE, using methanol or ethanol at pre-optimized concentration of 70:30 (*v*/*v*), to extract phenolic compounds from milled canola meal (0.75 mm) will be carried out at four different temperatures (140, 150, 160, and 170 °C) and five different time points (5, 10, 15, 20, and 30 min). This will establish the feasibility of MAE as a green and novel system for producing bioactives from canola meal.

## 2. Materials

Double expeller pressed canola meal containing oil content of 4–6% (*w*/*w*) (*Brassica napus* L.) was used as the substrate (Viterra group, St. Agathe, Manitoba). Sinapic acid (purity > 98%) was purchased from Fisher scientific Canada Ltd. (Ottawa, ON, Canada). Sinapine (purity > 97%) and canolol (purity > 98%) were purchased from ChemFaces Biochemical Co., Ltd. (Wuhan, Hubei, China). The extraction solvents were purchased from Fisher scientific Canada Ltd. (Ottawa, ON, Canada).

## 3. Methods

### 3.1. Sample Preparation

Canola meal was de-oiled using the Soxtec 2050 (Foss-Tecator, MN, USA) Khattab et al. [7], with few modifications. In brief, 15 g of canola meal sample was put into each extraction thimble and extracted with n-hexane with an optimized cycle of boiling, rinsing, and recovery for 30, 60, and 20 min, respectively. De-oiling was conducted for two consecutive cycles including all five replicates. At the end of the de-oiling process, the meal was separated, and the remaining oil was decanted. The de-oiled meal sample was milled using a SPEX™ SamplePrep 8000 M Mixer (Fisher scientific Canada Ltd., Ottawa, ON, Canada) to obtain a particle size of 0.75 mm. The size of the particles was confirmed via the Mastersizer 2000 (Malvern Instruments Ltd., Malvern, United Kingdom). The milled meal samples were stored at −20 °C until further analyzed.

### 3.2. Determination of Moisture Content

Moisture content of the defatted canola meal samples was conducted using a rapid method moisture meter (Denver instrument IR35, Denver, CL, USA). Samples were kept at 130 °C for 4 min to determine its moisture content. Ten replicates were analyzed, and the average moisture content was calculated to determine the phenolic content on dry weight basis.

### 3.3. Microwave-Assisted Solvent Extraction (MAE)

Microwave-assisted solvent extraction (MAE) of the defatted canola meal was conducted using the Multiwave^TM^ 5000 (Anton Paar, Montreal, QC, Canada) microwave system containing a rotor (20SVT50) with 20 vessels. Each vessel was filled with 2.0 g of defatted canola meal sample and extracted using 20.0 mL of 70% (*v*/*v*) methanol and 70% (*v*/*v*) ethanol. Prior to each extraction, a magnetic stirrer was used along with heated elements to evenly distribute the heat inside the vessel. The smart vent technology associated with the Multiwave^TM^ 5000 system ensured the proper maintenance of temperature and pressure throughout the experimental process. The power of the microwave system was kept at 1000 W, and during each extraction, the sample vessels were monitored for the changes in temperature using an IR temperature probe. The temperature calibration of the equipment was done prior to the extraction with the aid of water. The effect of MAE on extractability of phenolic compounds from the de-oiled canola meal and their antioxidant activity using several solvent-temperature-time combinations were evaluated. Ethanol and methanol were each used as extractants at four different temperatures (i.e., 140, 150, 160, and 170 °C) and five time intervals (i.e., 5, 10, 15, 20, and 30 min). Each solvent-temperature-time combination (40 in total) was replicated four times. After each extraction, the phenolic extract was taken out using plastic pasture tube and centrifuged at 7800× *g* for 15 min at 4 °C. The supernatant of the centrifuged samples was collected, and volume adjusted to 25.0 mL with the respective extraction solvents (methanol or ethanol) and kept at −20 °C until analyzed.

### 3.4. Identification of Major Sinapate Derivates Using HPLC-DAD

The changes in the phenolic composition of the extracts obtained with MAE were analyzed using high-performance liquid chromatography with diode array detection (HPLC-DAD) (Ultimate 3000; Dionex, Sunnyvale, Torrance, CA, USA) according to the method described by Nandasiri et al. [5]. The HPLC analysis was done using a reversed phased Kinetex Biphenyl C_18_ 100 Å RP column (2.6 µm, 150 × 4.6 mm, Phenomenex, Torrance, CA, USA) with a flow rate of 0.4 mL/min and 10 μL injection volume. The column was maintained at 30 °C for improved separation. The gradient consisted of formic acid (0.1%, *v*/*v*) in water as solvent A, and formic acid (0.1%, *v*/*v*) in methanol as solvent B. The gradient system was operated as follows: 25% B (0–3 min), 25–40% B (3–8 min), 40% B (8–13 min), 40–60% B (13–25 min), 60–70% B (25–38 min), 70–100% B (38–41 min), 100% B (41–44 min), 100–25% B (44–47 min), and 25% B (47–57 min). The chromatograms were acquired at both 320 nm (sinapine and sinapic acid) and 270 nm (canolol) using Chromeleon software Version 7.2 SR4 (Dionex Canada Ltd., Oakville, ON, Canada). Major sinapates were identified using the authentic standards with a detection limit of 0.001 mg/mL. Calibration curves for each standard were obtained from 1.0 to 100 μg/mL (n = 11) concentration range with R^2^ = 0.999 for sinapine, sinapic acid, and canolol.

### 3.5. Assessment of the Total Phenolic Content and Total Flavonoid Content

#### 3.5.1. Determination of Total Phenolic Content (TPC)

The TPC of the phenolic extracts obtained by MAE were determined using the Folin-Ciocàlteu assay as described by Thiyam et al. [19] with a few modifications. In brief, samples were diluted with distilled water with 1:100 (*v*/*v*) ratio. The diluted extract was mixed with 0.5 mL of Folin-Ciocalteu’s reagent and 1.0 mL of 19% (*v*/*v*) Na_2_CO_3_. The final volume was adjusted to 10 mL and the reaction mixture kept in dark for 60 min. The absorbance was measured at 750 nm using the UV-Visible Spectrometer FL6500 (Perkin Elmer Inc., Shelton, CT, USA). Methanol was substituted as blank, and sinapic acid solution (1.0 mM) was used to assemble the standard curve as presented in Appendix A.

#### 3.5.2. Determination of Total Flavonoid Contents (TFC)

The TFC of the obtained extracts were measured using the AlCl_3_ colorimetric method described by Zhishen et al. [20], with a few modifications. In brief, 0.5 mL of the extract was diluted in a ratio of 1:4 (*v*/*v*) with deionized water. The reaction mixture was prepared by adding 0.15 mL of NaNO_2_, 5% (*w*/*v*). After 6 min, 0.3 mL of AlCl_3_ 10% (*w*/*v*) was added to the reaction mixture and kept for additional 5 min, prior to adding 1.0 mL of NaOH (1 M). The absorbance was measured at 510 nm using the UV-Visible Spectrometer FL6500 (Perkin Elmer Inc., Shelton, CT, USA). Quercetin was used to prepare the standard curve (0.1 to 1 mM) Appendix A. TFC was expressed based on equivalent milligrams of quercetin per gram of dry weight of canola meal (QE mg/gDW).

### 3.6. Antioxidant Activity of the Phenolic Extracts Obtained by MAE

#### 3.6.1. 2,2-Diphenyl-1-Picrylhydrazyl (DPPH) Radical Scavenging Activity

The DPPH radical scavenging activity of the obtained extracts were determined according to the method of Girgih et al. [21], with slight modifications, using a 96-well micro plate reader (Bio-Tek Powerwave XS2, Winooski, VT, USA). Absorbance was measured at 517 nm wavelength, and the percentage radical scavenging activity was calculated using the following equation:DPPH radical scavenging activity (%) = (*Ab* − *As*/*Ab*) × 100 
where *Ab* and *As* are the respective absorbance of the blank and sample, respectively.

#### 3.6.2. Metal-ion Chelation Properties of the Extractants

The metal ion chelating ability was evaluated according to the modified method of Xie et al. [22] using a clear 96-well micro plate. The absorbance was measured at 562 nm using a microplate reader (Bio-Tek Powerwave XS2, Winooski, VT, USA). Methanol (70% *v*/*v*) was used as the blank and the results were expressed as a percentage of the metal ion chelating activity:% Metal ion chelating activity = (*Ab* − *As*/*Ab*) × 100 
where *Ab* and *As* are the absorbance of the blank and sample, respectively.

### 3.7. Statistical Analysis

All the experiments were carried out in four replicates. Results were presented as mean ± standard deviation of four replicate analysis. Data points were checked for their normality prior to the statistical analysis and required transformations were carried out to obtain normalized data [23]. To achieve the normalized data for the statistical model, square transformations were conducted [23]. For the current statistical analysis, different independent factors, including solvent (methanol, ethanol), temperature (140, 150, 160, and 170 °C), and time (5, 10, 15, 20 and 30 min), were assessed for the final concentration of the individual phenolic compounds, the major sinapates, and other unknown compounds. In addition, the relationship between the major sinapates and other unknown phenolic derivatives were determined using response surface methodology (RSM) (Appendix A).

The model fit statistics was conducted using the RSM analysis to obtain the best-fitting analysis. Over the years, the RSM technique was applied to obtain the best fitting model with the optimal response using minimal number of variables [24]. Furthermore, the RSM analysis provides complete information on the interaction effects between individual parameters for determining the stationary point which is the optimal condition [24]. Hence, to validate the proposed mathematical model created by the RSM analysis, an analysis of variance (ANOVA) is often required to assess the level of significance and model adequacy [25]. Statistical analysis was performed in R statistical software version 4.2.2 [26] using the packages ‘rsm’ [25] for RSM analysis and ‘corrplot’ [27] for producing correlation plots (Appendix A).

Similarly, the results of different antioxidant mechanisms were further assessed to determine the optimum extraction time/temperature combinations for the microwave-assisted solvent extraction.

## 4. Results and Discussion

### 4.1. Impact of Microwave-Assisted Solvent Extraction on the Major Sinapates

The impact of MAE was assessed to determine the changes in canola meal-derived endogenous polyphenolic compounds, such as sinapine, sinapic acid, and thermally generated canolol. These changes were assessed at the different time/temperature regimens used for both extractants, methanol (70%, *v*/*v*) and ethanol (70%, *v*/*v*). When subjected to microwave treatment, the major sinapates extracted increased with time and temperature reaching a maximum concentration prior to degradation. The thermally favored reactions involved in the conversion of sinapic acid to canolol and other thermo-generative compounds progressed over time [5,7]. A study conducted by Mayengbam et al. [28] indicated a 60% reduction in the original concentration of sinapine after roasting the canola seeds at 115 °C for 5 min, while the sinapine concentration further decreased to 90% after extraction for 20 min at 240 °C [28]. In a study conducted by Zago et al. [15], they reported that application of super-heated steam prior to microwave treatment increased the sinapine content of the meal fraction by 28%. This confirms that additional pre-treatments prior to MAE further facilitated the extractability of sinapic acid derivatives. Furthermore, Khattab et al. [7] reported that under the optimized condition, major phenolic compounds present in the meal were converted into sinapic acid, which increased its concentration from 0.14 to 10.2 mg/g.

The findings of the current study demonstrated that both extraction time and temperature significantly affected the extractability of the major sinapates (Figure 1). Furthermore, the two solvents produced different yields for the major sinapates with the MAE (Table 1). Previous reports found that MAE exhibited better extraction efficacy due to its synergetic effect on mass and heat transfer throughout the extraction process [29]. The yields obtained with the extractants depended on the composition of the extracting material, water content, solvent to substrate ratio, extraction time, and the temperature [29]. In addition, the intensity of the microwave also played a vital role in the extraction process. The intensity of the microwave is also recorded as the power density (W) per gram of sample. In the current study, the intensity was kept constant to minimize the variation throughout the extraction process. The solvent extractions conducted after the microwave-assisted pre-treatment showed that ethanol extracted higher amounts of sinapine compared to methanol (Table 1). RSM analysis between time and temperature on the concentration of major sinapates established the optimum extraction conditions for sinapine, sinapic acid, and canolol (Figure 1A,B). For both ethanol (adjusted R^2^—0.27) and methanol (adjusted R^2^—0.89) as extractants, only the main effects (time and temperature) had a significant effect on extractability (Table 2). The lower adjusted R^2^ value associated with ethanol may be due to the high variability of sinapine extractability at the relatively higher temperatures and prolonged extraction times. Furthermore, the statistical model indicated that there is no stationary point for the extraction of sinapine under the current extraction requirements using the microwave for both extractants. This was attributed to the longer processing times and conversion of sinapine into sinapic acid, canolol, and other sinapate derivatives during thermal processing [12,30,31]. It was further reported that the extractability of sinapine decreased with an increase in processing temperatures [32]. This was evident from the results of the ratio analysis between sinapine and sinapic acid, where the sinapine concentration had an inverse relationship with sinapic acid (Figure 2A,B).

In addition, our recent studies indicated that the conversion of sinapine to sinapic acid was higher compared to the conversion of sinapine/sinapic acid to canolol [11].

RSM analysis indicated that an extraction temperature of 126 °C for 33.84 min resulted in the highest conversion of sinapine to sinapic acid for methanol (adjusted R^2^—0.93), while 170 °C for 18.82 min (adjusted R^2^—0.62) was most effective for ethanol. The ratio analysis confirmed that methanol was a better extractant by facilitating the conversion of sinapine to sinapic acid at a lower temperature and time combination with the added benefit of lower energy costs (Figure 1A). Similar results were found in our previous studies, in which methanol and ethanol were used during accelerated solvent extraction (ASE) [5,11]. In addition, the higher adjusted R^2^ values for both extractants indicates that sinapine is the precursor for of sinapic acid. This was previously reported by Khattab et al. [6], in which sinapine could be converted to sinapic acid, sinapoyl glucose, and canolol. Moreover, the ratio analysis between sinapine and canolol also resulted in a higher adjusted R^2^ value for both methanol (adjusted R^2^—0.92) and ethanol (adjusted R^2^—0.75). These higher adjusted R^2^ values implies that the formation of canolol is dependent on sinapine as one of its precursors (Figure 1A,B).

Similarly, for sinapic acid with ethanol as the extractant, the main effects (time and temperature) had a significant impact on its extractability, although no stationary point was observed (Table 2). However, a stationary point at 163 °C at 16.18 min was observed for the extractant methanol. This showed that sinapic acid concentration increased with temperature and time, reaching an optimum at 163 °C with a processing time of 16.18 min (Table 2). Interestingly, the best response surface modeling observed for canolol with both extractants, although different stationary phases were recorded. For methanol, the stationary phase of canolol was at 151 °C with 15.43 min whereas, for ethanol its stationary point was located at 170 °C at 19.31 min (Table 1). Two different stationary points for each extractant further indicates that the extractability of canolol using the microwave can be optimized for each solvent. Based on the current results, methanol appears to be a better extractant compared to ethanol by using a lower processing time/temperature to generate canolol. Similar findings were reported by Khattab et al. [7] establishing the superiority of methanol as an extracting solvent for canolol. They also reported that around 95% of the total phenolics in canola meal were converted to sinapic acid with approximately 55% of sinapic acid decarboxylated to canolol under the microwave treatment. Another study conducted by Nandasiri et al. [33] reported that stationary point of response surface for canolol was located at 173.7 °C at 17.12 min for the 70% (*v*/*v*) methanol with the application of modified RapidOxy^®^ instrument in an inert environment. The lower processing temperature and time combinations reported to generate canolol using the MAE makes it an attractive method for use by the food industry (Table 3).

Based on the ratio analysis, it was evident that the conversion of sinapic acid to canolol had different values for both methanol and ethanol (Figure 2). For methanol, the stationary point for ratio analysis was at 159 °C with 10.89 min (adjusted R^2^—0.55), whereas for ethanol, it was at 170 °C with 17.63 min (adjusted R^2^—0.50). Consequently, methanol was the preferred medium for the conversion of sinapic acid to canolol, as it was more energy efficient. The adjusted R^2^ value around 0.5, however, indicates that sinapic acid was not the only precursor for the production of canolol.

### 4.2. Relationship among the Sinapates and Other Phenolic Derivatives

Apart from sinapine, sinapic acid, and canolol, nine other phenolic derivatives were observed with the microwave-aided solvent extraction at the different time/temperature combinations. Two different correlation plots were created for each extractant (Figure 3A,B). Strong, weak, positive, and negative correlations were evaluated using correlation plots. A positive correlation exists when two variables operate in unison, so that when one variable rises or falls, the other does the same. A negative correlation is when two variables move opposite one another so that when one variable rises, the other falls. The compounds with RT values are unknown phenolic compounds with interesting trends when thermally treated.

In terms of using methanol as extraction solvent, sinapine and sinapic acid had a significant and very strong positive correlation. A positive relationship was evident between sinapic acid and canolol in addition to sinapine and canolol. However, it was not significant (Figure 3A). Similar results were represented by Nandasiri et al. [33] and Khattab et al. [7], showing both sinapine and sinapic acid were precursors of canolol. Interestingly, sinapine had a significant negative relationship with unknown compounds, including RT-6.09, RT-8.21, RT-7.53, RT-10.10, and RT-13.66 (Figure 3A). These results suggest that these unknown compounds could be breakdown products of sinapine and sinapic acid. Of the unidentified compounds, RT-7.53, RT-10.10, and RT-13.66 were observed at 270 nm, while RT-6.09 and RT-8.21 were observed at a wavelength of 320 nm. A similar correlation pattern was also observed for sinapic acid with the abovementioned unidentified phenolic compounds (Figure 3A). This confirmed the strong positive correlation between sinapine and sinapic acid. The significant negative correlation between both sinapine and sinapic acid and the other phenolic compounds indicates the possibility that both sinapine and sinapic acid could be precursors for the generation of unknown phenolic compounds or degradation products of these major sinapates. A strong positive significant relationship was observed between sinapine and sinapic acid with the unknown compound of RT-21.36. Similar to sinapine and sinapic acid, this unknown RT-21.36 compound showed a negative relationship with RT-8.21, RT-7.53, RT-10.10, and RT-13.66 (Figure 3A). These results suggested that the unknown RT-21.36 compound could be a derivative of sinapine or sinapic acid. Moreover, the unknown compounds, including RT-6.09, RT-32.18, RT-8.21, RT-17.89, RT-7.53, RT-10.10, and RT-13.66, exhibited positive correlations among themselves, which shows that these compounds exhibit similar extractabilities among them, with methanol as the extraction solvent.

The same compounds with ethanol as the extractant demonstrated a quite different extractability for the unknown compounds. Of the identified compounds, RT-7.53, RT-10.10, and RT-13.66 were observed at 270 nm, while RT-6.09, RT-8.21, RT-14.46, RT-17.89, RT-21.36, and RT-31.18 were observed at a wavelength of 320 nm (Figure 3B). A strong and significant negative relationship was found between canolol and the unidentified compounds, RT-8.21, RT-14.46, and RT-17.89 (Figure 3B). This indicates that the concentration of canolol was impacted by these unidentified compounds. As canolol is highly reactive, we could assume that these compounds are degradation products of canolol. It was also found that these three unidentified compounds had a strong positive correlation between themselves. It appeared that these unidentified compounds may contribute to the formation or degradation of canolol. Similar to methanol, both sinapic acid and RT-21.36 had a significant negative relationship with the unknown RT-7.53 compound (Figure 3B). Furthermore, only the compounds RT-10.10 and RT-14.46 showed a negative correlation with the extractability of sinapine (Figure 3B). The unidentified compounds, RT-10.10 and RT-13.66, both showed a strong positive correlation, which further indicated that these two compounds showed similarities with extraction with ethanol.

### 4.3. Impact of MAE on Total Phenolic (TPC) and Total Flavonoid (TFC) Content

To determine the impact of MAE on the antioxidant activity of the phenolic extracts, two different antioxidant assays were used, each targeting a different mechanism. The first measured radical scavenging activity and the second assay determined the chelating ability of the metals. To understand the impact of MAE on its phenolic composition, both total phenolic (TPC) and total flavonoid contents (TFC) were assessed (Table 4). The results indicated that TPC of the extracts ranged from 36.87 ± 1.45 mg GAE/g DW to 117.75 ± 12.44 mg GAE/g DW (Table 4) depending on the extraction time-temperature regimes. Similarly, for the TFC the values ranged from 98.62 ± 0.50 mg QE/g DW to 333.59 ± 41.69 mg QE/g DW (Table 4).

A study conducted by Cong et al. [34] reported that microwave treatment could impact the TPC values significantly in different parts of rapeseed. They reported that the TPC in hull, cotyledon, endosperm, and entire rapeseed are 431, 1481, 1764, and 1395 mg/100 g, respectively, before the microwave treatment. However, after the microwave irradiation, these values changed to 698, 1447, 1695, and 1315 mg/100 g, respectively. Furthermore, Nandasiri et al. [5] reported that the application of accelerated solvent extraction (ASE) could impact the TPC levels differently in canola meal extracted with different extraction solvents. They reported that 70% ethanol extracts at 180 °C exhibited a TPC value of 24.71 ± 2.77 mg SAE/g, while 70% methanol extracts at 180 °C exhibited 20.72 ± 1.47 mg SAE/g. The current study found a similar trend with different results with 70% (*v*/*v*) ethanol being the preferred extraction solvent. Furthermore, Fadairo et al. [35] reported that a higher air-frying temperature increased the TPC. The optimum air-frying conditions reported were at 190 °C for 15 and 20 min with TPC values ranging from 3.15 ± 0.14 and 3.05 ± 0.02 mg GAE/g DW.

Furthermore, both total phenolic (TPC) and total flavonoid contents (TFC) of the samples were determined to assess the efficacy of MAE using different solvent systems with time*temperature regimes using a three-way ANOVA (Table 5). The results indicated that for TPC, all the major effects, including type of solvent, time, and temperature, had a significant effect (*p* < 0.05). Except for the time*temperature interaction, all other two-way and three-way interactions had a significant impact (*p* < 0.05) on the total phenolic content (Table 5). Statistical analysis further indicated that the total phenolic content was dependent on the type of solvent, time, and temperature and could be manipulated using these main effects. Interestingly, for TFC, only the main effects of type of solvent and temperature were significant (p < 0.05). This was also the case for TPC; except for the time*temperature interaction, all other two-way and three-way interactions had a significant impact (*p* < 0.05) (Table 5). The time factor was not significant, indicating that the extractability of TFC was independent of the duration of extraction.

### 4.4. Impact of MAE on the Antioxidant Activity

For both DPPH and MIC, the extraction time and type of solvent were not significant. Their activities were primarily dependent on extraction temperature. The results indicated that the DPPH activity of the methanol extracts ranged from 58.82 ± 3.88% (140 °C, 5 min) to 77.32 ± 2.06% (150 °C, 30 min) (Table 4). However, at both 170 °C, 20 min and 30 min also had higher radical activity for DPPH with 76.38 ± 1.40% and 76.38 ± 0.96%, respectively. In comparison, the methanol extracts ranged from 53.75 ± 3.69% (150 °C, 15 min) to 81.31 ± 0.18% (140 °C, 20 min) (Table 4). Furthermore, the time*temperature regime of 140 °C, 5 min also resulted in a radical scavenging activity of 80.93 ± 0.36% (Table 4). Metal ion chelating activity of the extracts showed a different trend compared to the DPPH radical activity. The lowest chelating activity percentage was observed at 170 °C, 20 min with 8.16 ± 1.09% for methanol extracts and 7.67 ± 1.14% (170 °C, 5 min) activity for the ethanol extracts (Table 4). Both extracts showed the highest inhibitory percentage at the temperature of 140 °C, pre-treatment temperature with 20.04 ± 0.81% at 10 min (methanol) and 27.38 ± 2.36% at 20 min (ethanol).

Similar studies on canola meal with microwave treatments by Sigar et al. [36] reported that the concentration of the lipophilic antioxidants increased with treatment time. They reported that the total antioxidant capacity of the extracts increased from 2 min of treatment time to 10 min from 0.84 to 5.83 mmol TEAC/L with the DPPH radical scavenging activity [36]. They also suggested that Maillard reaction products formed between carbohydrates and amino acids and amino phospholipids could account for a higher antioxidant activity. A study conducted by Yu et al. [37] stated that both steam explosion (270.12 ± 1.64 μmol/100 g oil) and microwave treatment (240.46 ± 3.87 μmol/100 g oil) yielded 100 times higher DPPH radical activity in rapeseed oil compared to the untreated oil (16.52 ± 1.18 μmol/100 g oil). Furthermore, another study conducted by Cong et al. [34] reported that DPPH scavenging activity of the hull, cotyledon, endosperm, and entire rapeseed are 642, 3813, 4128, and 3378 μmol TE/100 g sample, respectively. However, following a microwave treatment, DPPH scavenging activity changed to 902, 3716, 4177, and 3197 μmol TE/100 g sample. This study confirmed that the antioxidant activity was impacted with many processing factors.

Similarly, for TPC and TFC, except for time*temperature, all other two-way interactions had a significant impact (*p* < 0.05) on the DPPH radical scavenging activity (Table 5). The three-way interaction of solvent*time*temperature, however, had no significant impact on its antioxidant activity. Interestingly, in the case of MIC, only solvent*temp and time*temp interactions were significant (*p* < 0.05), except the solvent*time two-way interaction. Recent studies indicated that both sinapic acid and canolol showed higher radical scavenging activity. Higher radical scavenging activities are often closely associated with a reduction in cell oxidative stress [38]. Statistical analysis further indicated that, similar to DPPH, the three-way interaction of solvent*time*temperature was insignificant for the MIC activity of the extracts. The chelating power of the metal ions can be impacted by many factors including the geometry, ionic radii, valency, and hard-soft acid-base reflections [39]. Hence, in the current experiment the statistical results indicated that extraction temperature was the most important factor affecting the chelation power of the metals and its radical scavenging activity. Nevertheless, both types of solvent and extraction temperature were crucial factors for TPC and TFC.

### 4.5. Co-Relation Analysis of TPC, TFC and Antioxidnt Activity

The co-relation analysis between TPC, TFC, and antioxidant activity provided very interesting results. A strong and positive correlation was observed between TPC and TFC with antioxidant activity, as shown in Figure 4. This was further evidence of the ability of MAE to significantly increase both TPC and TFC levels. Flavonoids consist of many different classes, including anthocyanin, catechins, flavanone glycosides, flavanone, flavons, flavonol glycosides, flavonols, and isoflavons, are synthesized from the precursor phenyl alanine similar to most phenolic compounds [40]. Hence, the observed positive relationship could be explained. Furthermore, an increase in both TPC and TFC levels could be associated with heat-induced formation of novel phenolic compounds, including dimers, trimers, and other oligomers of sinapate derivatives and other flavor-active kaempferol derivatives [11]. Interestingly, no significant correlation was observed between MIC and DPPH (Figure 4), which further confirms the two different mechanisms of actions between the two antioxidant activities. Both DPPH and MIC showed a negative correlation with TPC. However, the correlation was not significantly different (Figure 4). Furthermore, DPPH radical scavenging activity showed a strong significant negative relationship with TFC. One of the limitations of the Folin-Ciocalteu assay is that it is based on colorimetry, and often the reaction could be reversible and facilitated by the presence of NH groups of the protein compounds [41]. Therefore, when it shows relatively higher TPC values, it could be due to the presence of other compounds. TPC also measures the reducing power of the extracts, and it is often recorded that there is a positive correlation between the TPC and the antioxidant activity. Hence, it is recommended to use different assays to measure the antioxidant activity of the samples [42].

TPC, similar to TFC, is also measured using a colorimetric assay based on the formation of a yellow-colored complex between the aluminum (Al^3+^) ion and the carbonyl and hydroxyl groups of flavonoids [40]. Some complexed flavonoid compounds show little or no antioxidant activity, which could explain the strong negative correlation between the TFC and the DPPH radical activity [40]. In addition, the antioxidant activity of DPPH is dependent on the formation of radicals [5]. With the more complexed and larger flavonoid molecules, the antioxidative radical scavenging activity could be limited to its structure–function relationship. Further analysis of more structure-based activity of antioxidants is required for confirmation of the above correlations.

## 5. Conclusions

MAE proved to be a novel and innovative green technique that is quick and requires much fewer solvents. This study was carried out using a Multiwave^TM^ 5000 microwave model provided by Anton Paar Ltd., Montreal, QC. Response surface methodology confirmed that the conversion of sinapine to sinapic acid and canolol was not only dependent on time and temperature, but also other factors, such as the solvent-substrate ratio. Correlation analysis showed that the extractability of sinapates was influenced by the type of solvent extractant, which further enhanced the yields of phenolic compounds and canolol. For antioxidant activity, extractant temperature appeared to be the most important factor, while the type of solvent extractant significantly affected the levels of TPC and TFC. Based on the results of this study, it is evident that MAE can be applied to the canola industry as a novel and reliable green method for extracting valuable phenolic antioxidants from canola meal.

## Figures and Tables

**Figure 1 foods-12-00318-f001:**
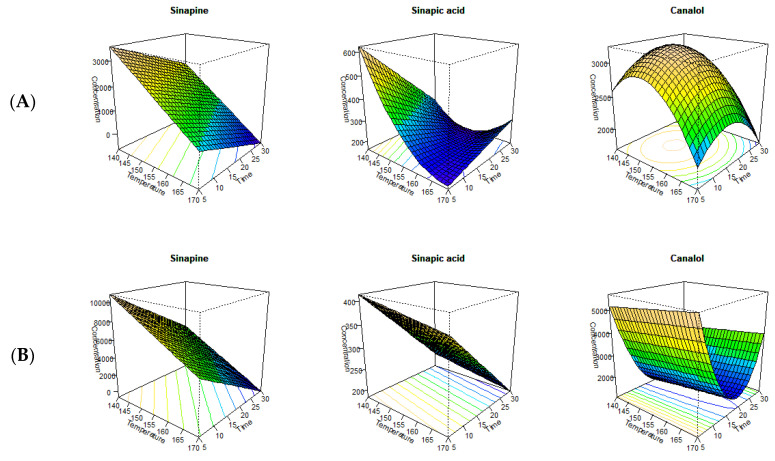
Response surface analysis of the major sinapates ((**A**)—methanol, (**B**)—ethanol).

**Figure 2 foods-12-00318-f002:**
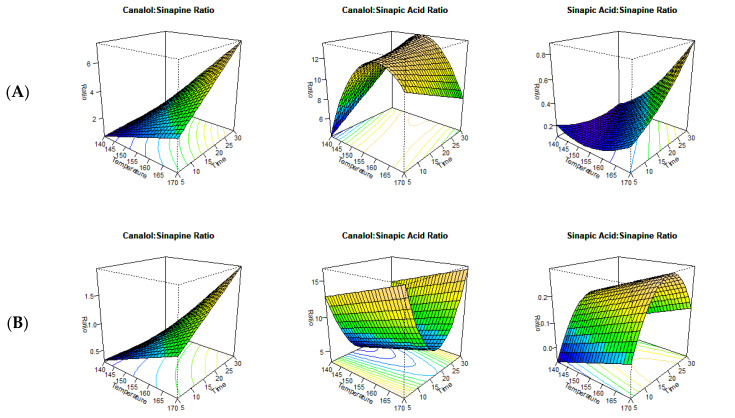
Ratio analysis of the major sinapates ((**A**)—methanol, (**B**)—ethanol).

**Figure 3 foods-12-00318-f003:**
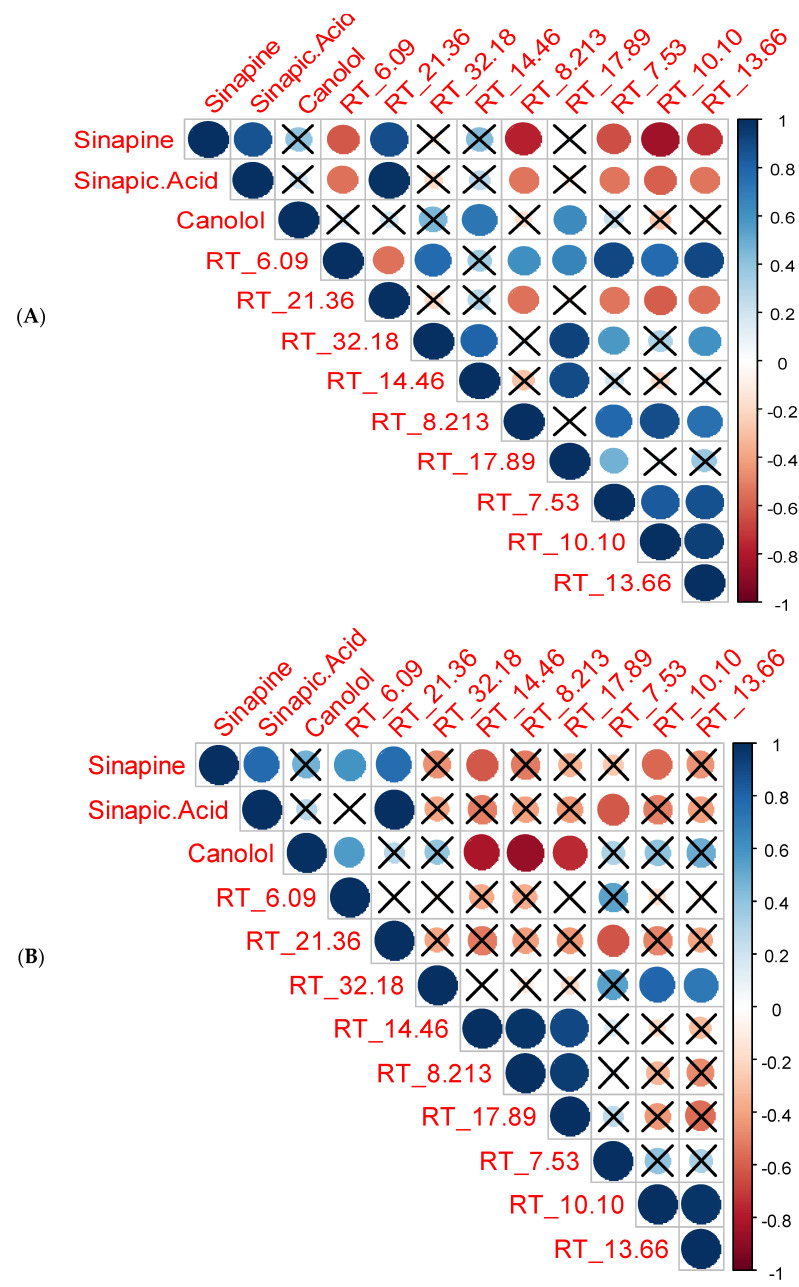
Correlation plot for the phenolic compounds ((**A**)—methanol, (**B**)—ethanol).

**Figure 4 foods-12-00318-f004:**
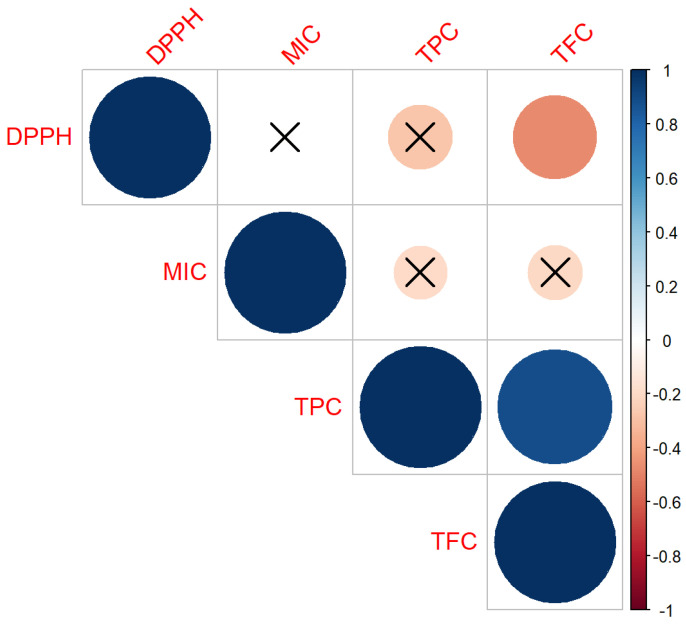
Correlation plot for the antioxidant activity.

**Table 1 foods-12-00318-t001:** Impact of MAE on the major sinapates.

Solvent	Temp (°C)	Time (min)	Wavelength (320 nm)								Wavelength (270 nm)			
			SP (μg/g DW)	SA (μg/g DW)	6.09 RT (μg SAE/g DW)	21.36 RT (μg SAE/g DW)	32.18 RT (μg SAE/g DW)	14.46 RT (μg SAE/g DW)	8.21 RT (μg SAE/g DW)	17.89 RT (μg SAE/g DW)	CL (μg/g DW)	7.53 RT (μg CLE/g DW)	10.10 RT (μg CLE/g DW)	13.66 RT (μg CLE/g DW)
Methanol	140	5	3541.83 ± 155.11	556.97 ± 57.17	181.56 ± 7.58	556.97 ± 57.17	4838.84 ± 118.08	258.23 ± 12.20	83.31 ± 4.11	210.78 ± 7.98	2343.56 ± 92.77	619.01 ± 36.52	287.41 ± 25.28	596.67 ± 15.14
		10	4046.75 ± 484.96	741.69 ± 120.51	222.07 ± 9.93	741.69 ± 120.51	6200.16 ± 191.57	318.07 ± 4.31	71.88 ± 2.54	258.27 ± 1.31	2983.46 ± 146.31	768.88 ± 38.90	356.14 ± 87.22	766.07 ± 12.28
		15	3158.83 ± 181.69	420.51 ± 88.93	243.20 ± 10.66	519.20 ± 88.61	6049.42 ± 111.09	297.49 ± 8.30	83.87 ± 9.45	269.85 ± 1.97	2847.28 ± 86.07	877.13 ± 29.00	349.37 ± 69.97	703.20 ± 25.17
		20	2872.39 ± 341.07	428.91 ± 97.34	231.71 ± 27.78	428.91 ± 97.34	6191.98 ± 401.07	308.62 ± 7.25	72.81 ± 2.78	256.73 ± 14.97	2852.00 ± 200.93	769.80 ± 29.59	302.71 ± 120.35	700.33 ± 53.17
		30	1759.47 ± 254.58	221.23 ± 20.80	314.12 ± 29.55	221.23 ± 20.80	7962.29 ± 253.80	341.19 ± 0.47	81.92 ± 6.38	308.68 ± 4.80	2547.76 ± 35.07	867.89 ± 51.68	760.42 ± 159.77	1208.22 ± 88.33
	150	5	2352.41 ± 84.48	334.21 ± 20.12	223.65 ± 13.72	334.21 ± 20.12	5759.54 ± 30.40	244. 98 ± 5.13	85.62 ± 4.05	232.79 ± 4.59	3006.33 ± 30.21	904.49 ± 30.85	720.48 ± 66.92	860.45 ± 63.07
		10	1744.25 ± 48.95	309.10 ± 32.83	229.32 ± 12.86	309.10 ± 32.83	5956.16 ± 123.09	270.60 ± 7.86	83.12 ± 2.15	248.94 ± 7.99	2965.80 ± 71.13	904.86 ± 34.47	762.96 ± 108.16	911.78 ± 105.42
		15	2585.63 ± 269.01	264.59 ± 34.52	301.83 ± 11.94	264.59 ± 34.52	6761.47 ± 599.85	292.47 ± 29.83	77.91 ± 4.01	287.09 ± 9.12	3158.14 ± 101.36	1123.46 ± 189.18	627.24 ± 18.76	803.68 ± 130.51
		20	1451.95 ± 40.07	190.18 ± 7.64	333.19 ± 16.55	187.49 ± 6.64	7202.72 ± 30.47	289.64 ± 8.31	86.40 ± 1.69	286.69 ± 2.65	3053.81 ± 20.02	1070.19 ± 64.62	1380.24 ± 132.61	1371.80 ± 79.54
		30	904.37 ± 32.75	234.61 ± 9.23	438.06 ± 10.38	237.57 ± 8.68	8257.27 ± 71.75	315.50 ± 2.29	94.77 ± 6.54	335.51 ± 7.92	2346.51 ± 59.78	1357.16 ± 44.19	2273.39 ± 224.10	1837.20 ± 35.43
	160	5	1411.93 ± 161.27	216.39 ± 4.72	315.38 ± 0.72	216.39 ± 4.72	6157.75 ± 1.91	281.13 ± 3.54	102.38 ± 4.50	271.40 ± 2.22	3202.70 ± 39.46	1171.92 ± 70.88	1695.63 ± 144.75	1314.73 ± 49.91
		10	1118.13 ± 147.94	232.74 ± 9.31	365.19 ± 10.23	232.74 ± 9.31	6770.70 ± 15.26	295.06 ± 14.93	112.07 ± 3.52	288.29 ± 4.52	3188.63 ± 29.96	1456.58 ± 67.43	2111.43 ± 238.84	1572.32 ± 84.69
		15	986.23 ± 20.44	237.16 ± 8.00	449.00 ± 25.90	237.16 ± 8.00	7740.05 ± 377.98	326.97 ± 17.39	129.50 ± 17.10	323.20 ± 9.95	3292.28 ± 50.80	1542.32 ± 172,48	2285.47 ± 534.93	1722.32 ± 248.64
		20	585.30 ± 110.11	208.61 ± 7.24	380.91 ± 26.83	208.61 ± 7.24	6648.04 ± 105.03	276.34 ± 6.83	129.90 ± 7.80	267.83 ± 5.37	2866.28 ± 39.30	1531.88 ± 89.23	2462.21 ± 506.36	1711.62 ± 189.13
		30	387.48 ± 11.25	232.77 ± 15.49	409.36 ± 12.25	232.77 ± 15.49	7459.99 ± 211.69	251.12 ± 11.77	109.86 ± 1.67	249.58 ± 11.04	1945.17 ± 32.29	1327.85 ± 7.83	3393.27 ± 151.29	2314.18 ± 64.22
	170	5	540.57 ± 68.50	186.63 ± 5.04	194.85 ± 16.41	186.63 ± 5.04	3455.70 ± 225.08	141.98 ± 9.18	100.59 ± 10.17	151.86 ± 12.34	1658.91 ± 93.44	719.88 ± 82.12	1279.74 ± 185.92	708.36 ± 103.34
		10	582.02 ± 169.18	219.38 ± 0.81	253.12 ± 35.32	219.38 ± 0.81	4599.25 ± 313.93	172.66 ± 3.20	112.85 ± 3.38	202.37 ± 1.75	1903.08 ± 38.79	855.28 ± 216.67	1931.11 ± 99.46	1065.82 ± 224.87
		15	481.70 ± 9.63	205.11 ± 2.60	319.09 ± 6.58	205.11 ± 2.60	5335.94 ± 139.30	195.41 ± 5.23	115.67 ± 12.72	209.67 ± 5.70	1985.13 ± 16.78	1276.08 ± 29.20	2166.43 ± 205.13	1464.37 ± 98.86
		20	421.36 ± 16.57	238.97 ± 3.86	377.30 ± 11.04	238.97 ± 3.86	6894.05 ± 120.57	258.06 ± 18.95	150.11 ± 1.33	250.05 ± 13.32	2610.34 ± 24.91	1636.74 ± 35.40	3642.72 ± 137.02	2169.27 ± 55.91
		30	ND	231.90 ± 7.32	322.13 ± 20.50	231.90 ± 7.32	8203.42 ± 275.74	221.97 ± 8.30	145.56 ± 1.80	187.77 ± 6.48	2049.56 ± 98.59	1464.48 ± 46.56	5593.86 ± 167.19	3050.06 ± 59.64
Ethanol	140	5	13394.58 ± 394.83	417.42 ± 11.00	629.55 ± 16.18	417.42 ± 11.00	138.50 ± 16.79	276.87 ± 10.45	110.81 ± 5.42	282.35 ± 2.56	4350.52 ± 54.57	1169.91 ± 49.43	1697.94 ± 132.17	1810.11 ± 110.97
		10	14044.18 ± 613.39	652.28 ± 58.64	565.26 ± 83.13	652.28 ± 58.64	109.28 ± 109.09	238.00 ± 18.34	90.48 ± 15.66	265.95 ± 26.55	4169.30 ± 139.70	950.81 ± 179.61	1021.11 ± 272.01	1523.05 ± 170.04
		15	1095.05 ± 95.90	170.76 ± 1.85	324.53 ± 12.22	170.76 ± 1.85	125.16 ±19.93	750.63 ± 21.13	2122.54 ± 68.84	859.48 ± 34.65	538.81 ± 27.04	1525.70 ± 84.35	328.36 ± 22.39	430.76 ± 10.36
		20	731.83 ± 28.05	142.07 ± 1.57	261.80 ± 37.63	142.07 ± 1.57	66.71 ± 1.79	611.53 ± 9.80	2226.05 ± 31.34	514.25 ± 25.58	270.67 ± 33.72	946.23 ± 93.50	nd	377.84 ± 73.87
		30	10327.27 ± 598.03	258.08 ± 33.51	704.34 ± 35.90	258.08 ± 33.51	69.61 ± 0.58	246.42 ± 14.81	125.14 ± 22.66	328.16 ± 24.40	3760.24 ± 23.55	1507.96 ± 154.60	1953.09 ± 351.98	1811.69 ± 243.16
	150	5	13340.43 ± 466.82	668.37 ± 72.76	336.87 ± 28.19	668.37 ± 72.76	65.79 ± 5.48	202.75 ± 9.94	54.52 ± 4.95	162.75 ± 12.37	3918.48 ± 70.76	512.99 ± 35.21	445.64 ± 68.75	1091.00 ± 163.78
		10	11928.58 ± 587.55	313.74 ± 14.92	825.75 ± 17.45	313.74 ± 14.92	179.26 ± 2.60	297.90 ± 8.31	159.14 ± 6.41	359.33 ± 3.36	5029.25 ± 102.36	1794.05 ± 119.72	2778.22 ± 360.85	2520.22 ± 164.79
		15	878.97 ± 40.82	108.69 ± 34.34	289.28 ± 16.68	108.69 ± 34.34	146.69 ± 1.05	678.77 ± 3.57	1965.64 ± 241.53	855.42 ± 19.14	852.33 ± 70.13	1480.60 ± 504.98	nd	455.58 ± 75.95
		20	939.93 ± 14.51	247.26 ± 8.24	273.74 ± 40.11	247.26 ± 8.24	181.81 ± 6.89	654.99 ± 31.13	1716.51 ± 19.07	515.48 ± 2.17	737.24 ± 23.34	1943.03 ± 133.25	nd	644.68 ± 65.59
		30	6785.0 ± 538.36	265.40 ± 9.83	638.69 ± 26.42	265.40 ± 9.83	224.89 ± 13.76	258.31 ± 14.98	176.61 ± 4.25	336.76 ± 8.87	4282.47 ± 84.33	1908.96 ± 30.18	3764.29 ± 165.32	2834.63 ± 73.84
	160	5	8302.35 ± 860.25	328.12 ± 22.21	631.72 ± 15.45	328.12 ± 22.21	261.06 ± 72.65	272.09 ± 42.43	199.40 ± 12.95	295.88 ± 14.10	5404.57 ± 124.98	2095.57 ± 104.81	4546.45 ± 685.77	2999.54 ± 270.58
		10	8272.26 ± 840.31	302.36 ± 2.55	684.68 ± 49.72	302.36 ± 2.55	243.06 ± 10.69	305.89 ± 9.28	185.33 ± 5.23	291.94 ± 4.67	4734.04 ± 11.84	2081.81 ± 58.34	3909.07 ± 344.63	2857.57 ± 123.19
		15	1094.40 ± 28.28	327.38 ± 12.44	254.10 ± 18.16	327.38 ± 12.44	146.25 ± 0.48	504.83 ± 36.63	1834.48 ± 20.88	581.44 ± 17.46	1560.00 ± 37.14	2821.81 ± 58.34	nd	821.35 ± 38.38
		20	944.80 ± 20.99	294.51 ± 27.21	285.19 ± 12.46	312.22 ± 28.53	189.17 ± 7.11	470.56 ± 12.64	1150.89 ± 66.09	478.56 ± 8.09	894.85 ± 25.89	3747.86 ± 241.71	nd	1034.51 ± 98.33
		30	3854.23 ± 599.14	238.82 ± 9.57	334.72 ± 21.76	238.82 ± 9.57	235.23 ± 14.66	278.73 ± 15.50	184.71 ± 12.07	210.10 ± 8.44	3569.89 ± 7.25	1175.90 ± 52.81	5148.58 ± 679.12	3380.19 ± 294.13
	170	5	5341.16 ± 379.86	301.95 ± 5.04	463.19 ± 8.19	301.95 ± 5.04	470.52 ± 15.76	329.48 ± 44.20	223.39 ± 28.94	258.83 ± 11.57	5003.13 ± 249.93	1831.71 ± 79.53	5920.86 ± 554.38	3539.47 ± 294.13
		10	3245.36 ± 382.16	276.55 ± 42.98	412.01 ± 67.52	276.55 ± 42.98	216.79 ± 29.15	326.69 ± 28.49	202.20 ± 4.00	240.26 ± 22.37	4296.95 ± 126.97	1625.46 ± 153.06	5404.53 ± 499.06	3334.62 ± 315.11
		15	937.75 ± 69.16	394.65 ± 20.76	287.31 ± 4.83	394.65 ± 20.76	146.25 ± 0.48	523.75 ± 26.06	1379.06 ± 188.27	530.22 ± 21.78	1504.42 ± 39.98	3151.65 ± 88.78	nd	923.38 ± 51.81
		20	855.03 ± 49.93	267.87 ± 28.44	282.82 ± 6.53	298.73 ± 34.15	214.01 ± 6.75	406.18 ± 18.74	647.17 ± 198.94	289.21 ± 9.86	1066.19 ± 27.66	4154.29 ± 328.15	nd	1058.08 ± 65.12
		30	1504.51 ± 65.14	231.89 ± 15.70	295.98 ± 15.16	253.84 ± 28.15	239.13 ± 3.88	347.94 ± 19.44	237.64 ± 7.91	210.80 ± 13.28	3478.94 ± 80.29	1088.08 ± 67.76	6447.80 ± 457.07	4502.06 ± 263.64

SP: sinapine, SA: sinapic acid, CL: canolol, temp: temperature, SAE: sinapic acid equivalents, CLE: canolol equivalents, min: minutes, RT: retention time, DW: dry weight, nm: nanometer, μg: microgram, g: gram, nd: not detected.

**Table 2 foods-12-00318-t002:** Response surface analysis of optimized conditions for major sinapates.

	RSM Parameters	Estimate	STD Error	*t*-Value	Level of Significance
Methanol	Sinapine				
	Time	−283.68	54.26	−5.23	0.00 *
	Temp	−373.24	32.57	−11.46	0.00 *
	R^2^—0.9010				
	Adj R^2^—0.8886				
	Sinapic Acid				
	Time	21.10	14.45	1.46	0.16
	Temp	60.90	20.52	2.97	0.01 *
	Time * Temp	12.26	3.09	3.97	0.00 *
	Temp^2^	7.85	2.38	3.30	0.01 *
	R^2^—0.8273				
	Adj R^2^—0.7812				
	Canolol				
	Time	−430.74	132.88	−3.24	0.01 *
	Temp	−395.77	87.28	−4.53	0.00 *
	Time^2^	−73.92	25.73	−2.87	0.01 *
	Temp^2^	−41.96	11.15	−3.76	0.00 *
	R^2^—0.7043				
	Adj R^2^—0.6255				
Ethanol	Sinapine				
	Time	−1164.29	545.42	−2.14	0.05 *
	Temp	−756.26	335.72	−2.25	0.04 *
	R^2^—0.3617				
	Adj R^2^—0.2866				
	Sinapic Acid				
	Time	−38.58	16.80	−2.30	0.04 *
	Temp	−4.92	10.34	−0.48	0.64
	R^2^—0.2444				
	Adj R^2^—0.1555				
	Canolol				
	Time	2162.86	435.07	4.97	0.00 *
	Temp	64.99	81.65	0.80	0.44
	Time^2^	505.72	84.24	6.00	0.00 *
	R^2^—0.727				
	Adj R^2^—0.676				

* significant at the level of 0.05; STD, standard; Temp, Temperature; RSM, response surface methodology analysis; R^2^, coefficient of correlation; Adj R^2^, adjusted coefficient of correlation.

**Table 3 foods-12-00318-t003:** Analysis of variance (ANOVA) table for the major sinapates.

		DF	Sum Sq	Mean Sq	F Value	Level of Significance
Methanol	Sinapine					
	FO (Time * Temp)	2	21463937	10731968	72.77	0.00 *
	Residuals	16	2359552	147472		
	Lack of fit	16	2359552	147472		
	Pure error	0	0			
	Sinapic Acid					
	FO (Time * Temp)	2	199644	99822	22.60	0.00 *
	TWI (Time * Temp)	1	69559	69559	15.75	0.00 *
	PQ (Temp)	1	48112	48112	10.89	0.01 *
	Residuals	15	66245	4416		
	Lack of fit	15	66245	4416		
	Pure error	0	0			
	Canolol					
	FO (Time * Temp)	2	1294540	647270	6.66	0.01 *
	PQ (Time * Temp)	2	2177360	1088680	11.20	0.00 *
	Residuals	15	1457565	97171		
	Lack of fit	15	1457565	97171		
	Pure error	0	0			
Ethanol	Sinapine					
	FO (Time * Temp)	2	169613846	84806923	4.82	0.02 *
	Residuals	17	299383971	17610822		
	Lack of fit	17	299383971	17610822		
	Pure error	0	0			
	Sinapic Acid					
	FO (Time * Temp)	2	91906	45953	2.75	0.09
	Residuals	17	284183	16717		
	Lack of fit	17	284183	16717		
	Pure error	0	0			
	Canolol					
	FO (Time * Temp)	2	6902440	3451220	3.31	0.06
	PQ (Time)	1	37546925	37546925	36.04	0.00 *
	Residuals	16	16667785	1041737		
	Lack of fit	16	16667785	1041737		
	Pure error	0	0			

* significant at the level of 0.05; DF, degrees of freedom; Temp, Temperature; Sum Sq, sum of squares; mean sq, mean sum of squares, F-value; FO, first-order response surface (i.e., linear function); TWI, two-way interaction; PQ, pure quadratic terms.

**Table 4 foods-12-00318-t004:** Assessment of the total phenolic content (TPC)/total flavonoid content (TFC) and antioxidant activity.

Solvent	Temperature °C	Time (min)	TPC (mg SAE/g DW)			TFC (mg QE/g DW)			DPPH Activity (%)			Metal Ion Chelation (%)		
Methanol	140	5	47.32	±	0.44	115.18	±	2.43	58.82	±	3.88	12.52	±	0.54
		10	36.88	±	1.45	140.62	±	3.97	73.15	±	1.16	20.04	±	0.81
		15	47.11	±	2.00	134.86	±	6.11	64.10	±	2.99	16.66	±	0.14
		20	43.49	±	2.20	125.99	±	7.40	73.44	±	1.77	13.18	±	1.47
		30	44.05	±	2.85	134.99	±	6.85	73.78	±	0.63	16.58	±	0.04
	150	5	47.93	±	1.11	141.50	±	9.50	68.68	±	0.49	15.66	±	0.77
		10	46.72	±	1.94	140.27	±	5.46	66.70	±	2.17	17.04	±	0.53
		15	49.17	±	5.21	134.41	±	17.51	70.00	±	1.77	19.10	±	1.24
		20	58.88	±	2.80	156.45	±	8.87	72.42	±	1.33	12.01	±	1.00
		30	65.61	±	1.16	167.44	±	6.43	77.32	±	2.06	16.79	±	1.76
	160	5	64.63	±	1.03	157.99	±	8.23	73.15	±	1.88	11.23	±	0.64
		10	68.68	±	1.13	158.23	±	10.43	73.78	±	1.60	11.62	±	0.36
		15	73.72	±	2.69	187.31	±	18.71	76.17	±	2.86	11.39	±	1.05
		20	64.88	±	1.98	147.76	±	7.40	74.09	±	0.63	11.29	±	0.99
		30	63.90	±	1.06	151.78	±	4.60	73.28	±	0.22	11.75	±	0.71
	170	5	57.97	±	4.85	124.98	±	9.30	66.08	±	1.09	8.92	±	0.30
		10	57.75	±	2.41	143.43	±	8.42	72.81	±	0.44	8.49	±	0.46
		15	67.65	±	1.61	156.28	±	11.50	74.19	±	1.49	9.16	±	0.63
		20	84.87	±	1.61	194.98	±	8.04	76.38	±	1.40	8.16	±	1.09
		30	96.27	±	3.02	230.38	±	3.82	76.38	±	0.96	14.59	±	1.97
Ethanol	140	5	41.30	±	1.16	133.21	±	3.56	80.93	±	0.36	11.15	±	0.11
		10	66.64	±	2.35	155.60	±	17.99	79.98	±	0.73	15.83	±	0.82
		15	54.14	±	2.29	161.64	±	2.40	78.37	±	0.43	14.29	±	0.47
		20	78.46	±	4.51	98.62	±	0.50	81.31	±	0.18	27.38	±	2.36
		30	68.10	±	1.99	137.47	±	4.11	77.52	±	1.31	12.37	±	0.61
	150	5	65.85	±	2.42	150.25	±	3.70	78.93	±	0.36	12.23	±	0.36
		10	86.17	±	2.30	199.86	±	11.97	74.97	±	2.51	16.32	±	1.82
		15	48.28	±	4.93	168.71	±	32.05	53.75	±	3.69	20.72	±	0.03
		20	67.25	±	3.08	122.34	±	14.48	79.12	±	0.95	19.63	±	1.11
		30	92.27	±	1.41	222.12	±	16.42	71.43	±	1.28	10.40	±	0.11
	160	5	112.27	±	12.24	306.87	±	53.08	68.21	±	1.19	10.01	±	0.48
		10	105.91	±	2.93	279.59	±	28.91	68.88	±	3.09	9.52	±	1.18
		15	105.44	±	1.70	307.32	±	19.44	66.02	±	0.88	14.29	±	0.05
		20	93.72	±	0.88	212.27	±	18.62	68.41	±	1.63	9.77	±	1.59
		30	96.84	±	2.78	228.23	±	10.47	64.10	±	5.08	11.28	±	1.82
	170	5	117.75	±	12.44	333.59	±	41.69	61.63	±	3.33	7.67	±	1.14
		10	110.96	±	4.04	318.57	±	26.04	61.78	±	0.72	16.95	±	0.47
		15	106.44	±	1.05	322.82	±	23.84	62.02	±	3.41	19.25	±	0.57
		20	82.33	±	8.12	165.65	±	32.36	70.07	±	4.87	16.53	±	1.35
		30	109.85	±	1.87	277.09	±	26.19	56.65	±	2.42	16.35	±	2.69

TPC, total phenolic content; TFC, total flavanoid content; SAE, sinapic acid equivalents; QE, quercetin equivalents; EDTA, ethylenediaminetetraacetic acid; DW, dry weight; mg, milligram; g, gram; min, minutes; °C, centigrade.

**Table 5 foods-12-00318-t005:** Three-way Analysis of variance (ANOVA) table for the Antioxidant Activity.

		DF	Sum Sq	Mean Sq	F Value	Level of Significance
TPC	Solvent	1	42092	42092	416.67	0.00 *
	Time	1	2402	2402	23.78	0.00 *
	Temp	1	52303	52303	517.75	0.00 *
	Solvent * Temp	1	3299	3299	32.66	0.00 *
	Solvent * Time	1	1015	1015	10.05	0.00 *
	Time * Temp	1	95	95	0.95	0.33
	Solvent * Time * Temp	1	5306	5306	52.53	0.00 *
	Residuals	228	23033	101		
TFC	Solvent	1	158372	158372	193.82	0.00 *
	Time	1	1242	1242	72.77	0.22
	Temp	1	207308	207308	253.72	0.00 *
	Solvent * Temp	1	13562	13562	16.60	0.00 *
	Solvent * Time	1	91400	91400	111.86	0.00 *
	Time * Temp	1	1	1	0.00	0.97
	Solvent * Time * Temp	1	15322	15322	18.75	0.00 *
	Residuals	139	113575	817		
DPPH	Solvent	1	13.14	13.14	1.03	0.31
	Time	1	33.64	33.64	2.64	0.11
	Temp	1	772.06	772.06	60.55	0.00 *
	Solvent * Time	1	413.09	413.09	32.40	0.00 *
	Solvent * Temp	1	2000.75	2000.75	156.91	0.00 *
	Time * Temp	1	4.43	4.43	0.347	0.56
	Solvent * Time * Temp	1	6.28	6.28	0.49	0.48
	Residuals	103	1313.39	12.75		
MIC	Solvent	1	4.41	4.41	0.43	0.52
	Time	1	9.65	9.65	0.93	0.34
	Temp	1	123.76	123.76	11.96	0.00 *
	Solvent*Time	1	1.03	1.03	0.10	0.75
	Solvent*Temp	1	138.92	138.92	13.42	0.00 *
	Time*Temp	1	41.51	41.51	4.01	0.05 *
	Solvent*Time*Temp	1	0.87	0.87	0.08	0.77
	Residuals	76	786.73	10.35		

* Significant at the level of 0.05; DF, degrees of freedom; Temp, Temperature; Sum Sq, sum of squares; Mean sq, mean sum of squares, F-value; TPC, total phenolic content; TFC, total flavonoid content; MIC, metal ion chelation activity; DPPH, 2,2-Diphenyl-1-picrylhydrazyl radical scavenging activity.

## Data Availability

The datasets generated for this study are available on request to the corresponding author.

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
