# Peer review of "Optimization of Canolol Production from Canola Meal Using Microwave Digestion as a Pre-Treatment Method"

_foods, 2023, doi:10.3390/foods12020318_

Round 1

Reviewer 1 Report

Reviewers' comments:

Manuscript Number: foods-2072868

Title: Optimized Production of Canolol Using Microwave Digestion as a Method of Pre-Treatment
The MS deals with the application of using Microwave Technology for the extraction of Canolol under optimized condition. The work is interesting since it is reporting the application of Microwave-assisted solvent extraction as a green extraction method in industrial sectors. However, I have several concerns which needs to be addressed, my comments and suggestions are listed below:

English editing is needed for this manuscript. Several statements should be re-written, Authors have to carefully check the English grammar throughout the manuscript.

Comments

-          Line 167-169: In Conclusion: “The results confirmed that conversion of sinapine to sinapic acid and canolol is not only dependent on time and temperature but other intrinsic and extrinsic factors”. Please revise this sentence

-          Line 266-267: “The impact of MAE was conducted to determine the compositional changes in sinapine, sinapic acid and thermally generated canolol”. Please check and revise the sentence.

-          Authors should improve the Results and discussion section by explaining and comparing the quantitative data (eg., line 285-286: furthermore, the two solvents produced different yields for the major sinapates with the MAE) and compare with the previous literature. For eg., 10.1007/s11947-017-1934-z cite and compare with the literature to improve the manuscript.

-          Author should provide the details of different variables and responses for better understanding.

-          Author should give the separate section about the RSM experimental design of this study in the materials and methods section.

-          There is lack of discussions and relevant recent references in the section 4, Include more relevant references and discuss, elaborately.

-          Conclusion: author should revise the conclusion section and highlight the important findings with the data.

-          Line 137 & 145: Please always leave a space between number and SI unit (e.g. 2.0 g, 130 ℃), however no space before the "%", "/" and ":" signs. 

Author should provide the RSM software and version used in this study

Author Response

Dear Reviewer,

Thank you for giving me the opportunity to submit a revised version of our manuscript titled “Optimization of Canolol Production from Canola Meal Using Microwave Digestion as a Pre-Treatment Method’’.

We sincerely appreciate the time and effort that you have devoted to providing your insightful feedback/comments on the manuscript. We have been able to address and incorporate changes to reflect most of the suggestions/concerns. Detailed responses to the concerns are attached in the word document.

Thank you

Reviewer 2 Report

The article entitled “Optimized Production of Canolol Using Microwave Digestion as a Method of Pre-Treatment” is a study on the use of microwave heating for an optimized extraction of canolol from de-oiled canola products. This work performed various studies to come to a conclusion. However, it has following major issues to be addressed before it is ready for publication in Foods.

1.     The title could be more comprehensive and clearer; as for example, the authors could include the source (i.e., raw material), from which canolol was extracted?

2.     Abstract (Line 20-21): “at the same time” repeated two times in the same sentence?

3.     Line 27: “Three antioxidant assays” – There are only two antioxidant assay methods mentioned in M&M section?

4.     Line 40: “In Chapter 3 of their 2022” – does not give a clear enough meaning?

5.     Line 62: “……. canolol contributes to protect lipids and proteins of oxidation” – please remove “of”?

6.     Line 64: “……..have been shown to exhibit illustrate” – two verbs together; please delete one of them?

7.     Line 66-74: This portion is related to oil extraction, not the phenolic compounds extraction. Since this work is related to phenolic extraction from defatted canola meal, this portion can be replaced by the information on solvent extraction using different solvents?

8.     Line 88: “thermos-generative” may be “thermo-generative”?

9.     Line 95: “temperatures”- please insert a full stop (.) after this?

10.  Line 123-124: “Canola meal was de-oiled using Soxtec 2050 (Foss -Tecator, Foss 124 North America, MN, USA) following the method of Khattab et al. [6] with slight modifications.”

11.  Line 126: “……30, 60 and 20 min” – which solvent was used for oil extraction? Also, which solvent boils at 30 and 20 oC?

12.  Line 130: please provide the company name of ball mill? Please confirm the powder particle size of 0.75 mm?

13.  Line 152: “sample vassals” – please correct “vassals” as “vessels”?

14.  Line 157: “7800 g” – please correct as “7,800× g”.

15.  Line 167: “Kinetex Biphenyl C18 100 Å RP column (2.6 mm, 150 × 4.6 mm,” – please check the specification of the column; probably 2.6 μm, 150 mm × 4.6 mm)?

16.  Line 170: “elution of formic acid (0.1%, v/v) in water as solvent A and formic acid (0.01%, v/v) in methanol as solvent B”?

17.  Line 173: “70% - 100% B” and “100% B” – you missed “B”?

18.  Line 174: “100% - 25% B”

19.  Line 177: “Major phenolic compound canolol was identified using the authentic standards……” – why only canolol? How about other two compounds – sinapic acid and sinapine? Please mention the remaining two compounds as well.

20.  Line 180-181: “concentration range with R2 = 0.999 for all the three compounds (sinapic acid, canolol, and sinapine).

21.  There should be a space (gap) between a number and a unit (for example, 170 oC; 1 mL; 2.0 g etc.) – please check the whole document for such corrections.

22.  In case of “p<0.05” or “p>0.05”, p should be italic (p). For a significant effect, “p<0.05”; not “p>0.05”

23.  Line 297: (Table 3) – Table 1 and Table 2 should appear before Table 3 in the manuscript.

24.  Line 311-312: “This was evident from the results of the ratio analysis between sinapine and sinapic acid (Figure 3).” - Please explain in brief from Fig. 3 regarding the high/low ratio of these compounds?

25.  Line 318: “Figure 3” – Figure 2 should come before Figure 3.

26.  Table 3: Table 1 and Table 2 should appear before Table 3.

27.  Table 3: Table presentation is too messy, especially in the second and third parts, columns are not properly located.

28.  Table 1: WHY Table 1 is behind Table 3?; Why these compounds (Sinapine, sinapic acid, canolol) for Methanol and Ethanol have different parameters? For example, in case of Methanol - Sinapine (Time, and Temp); Sinapic acid (Time, Temp, Time*Temp, and Temp2); Canolol (Time, Temp, Time2, and Temp2). In case of Ethanol -  Sinapine (Time, and Temp); Sinapic acid (Time, and Temp); Canolol (Time, Temp, and Time2).

29.  Figure 2a & b (Line 46): Can you please explain a bit about this positive and negative correlation results - what is the meaning of a negative or a positive correlation between two compounds? Rather, a correlation plot among the compounds detected and operating parameters (time, temp) would give interesting results? In the manuscript, a and b; but in Figure 2, A and B? What these unknown compounds with various RT values - Are they phenolic compounds or something else?

30.  Table 1b (Footnote): FO, TWI, PQ – what these abbreviations mean?

31.  Section 4.3 (Line 90): “three different antioxidant assays…..” this manuscript talks only two, such as DPPH and metal chelating ability?

32.  Section 4.3 (Line 94): How much TPC and TFC were detected? Before a statistical analysis of these results, please provide the results of TPC and TFC.

33.  Moreover, TPC and TFC are not antioxidant activities? Please make a separate section with a heading like "Impact of MAE on the TPC and TFC" or similar one!

34.  Section 4.3 (Line 103): In Table 2, “time:temp”? Same issues for other interacting parameters in Table 2.

35.  Section 4.3 (Line 111): Before a statistical analysis of DPPH and MIC, please provide the results obtained using different extraction protocols.

36.  Section 4.3 (Line 123-125): This work determined the antioxidant activity of the phenolic compounds in canola meal, not about the metal ions; so, this reference 30 is irrelevant (In my opinion.)?

37.  Section 4.3 (Line 150-162): Please revise this section thoroughly to make it more relevant to this study.

38.  Conclusion: Please re-write conclusion section based on the results reported; for example, there is no results / study about “less solvent consumption” and “shorter time” reported in Results and Discussion section.

39.  Figure S1A and Figure S1B: please correct “using” as “using”?

40.  English language revision is necessary; many typos are seen, which need correction.

41.  Discussion of the results obtained is poorly done; therefore, discussion of each result, based on the reported result(s) is required and recommended.

Author Response

(The authors gave the same response as above.)

Round 2

Reviewer 2 Report

The authors have taken proper action against all of previous comments for the issues raised against the manuscript no. foods-2072868. Thank you!

Line 115-118: The values with two digits (maximum) after decimal may be enough to express these results. Also, the values are normally presented without standard deviation in the text (In tables only, Mean value ± SD). For example, 36.87 mg GAE/g DW, instead of 36.8752 ± 1.4503 mg GAE/g DW (Line 115). This applies to all the values presented in the manuscript.

Author Response

Thank you for your comment. We have updated the document.

“The results indicated that the TPC of the extracts ranged from 36.87 ± 1.45 mg GAE/g DW to 117.75 ± 12.44 mg GAE/g DW (Table 3) depending on the extraction time-temperature regimes. Similarly, for the TFC the values ranged from 98.62 ± 0.50 mg QE/g DW to 333.59 ± 41.69 mg QE/g DW (Table 3).”